# Electrochemical Sensors for Detection of Markers on Tumor Cells

**DOI:** 10.3390/ijms22158184

**Published:** 2021-07-30

**Authors:** Han Zhou, Xin Du, Zhenguo Zhang

**Affiliations:** 1Shandong Provincial Key Laboratory of Animal Resistance Biology, College of Life Sciences, Shandong Normal University, Jinan 250014, China; zairvan@163.com; 2Key Laboratory of Food Nutrition and Safety, College of Life Sciences, Shandong Normal University, Jinan 250014, China

**Keywords:** tumor cell, electrochemical, biosensor, probe, marker

## Abstract

In recent years, the increasing incidence and mortality of cancer have inspired the development of accurate and rapid early diagnosis methods in order to successfully cure cancer; however, conventional methods used for detecting tumor cells, including histopathological and immunological methods, often involve complex operation processes, high analytical costs, and high false positive rates, in addition to requiring experienced personnel. With the rapid emergence of sensing techniques, electrochemical cytosensors have attracted wide attention in the field of tumor cell detection because of their advantages, such as their high sensitivity, simple equipment, and low cost. These cytosensors are not only able to differentiate tumor cells from normal cells, but can also allow targeted protein detection of tumor cells. In this review, the research achievements of various electrochemical cytosensors for tumor cell detection reported in the past five years are reviewed, including the structures, detection ranges, and detection limits of the cytosensors. Certain trends and prospects related to the electrochemical cytosensors are also discussed.

## 1. Introduction

As some of the most threatening diseases in the world, different kinds of cancers display high morbidity and mortality rates (Figure 1) [1]. To ensure successful treatment, it is essential to monitor tumor cells rapidly, accurately, and sensitively. To cure cancers successfully, we require more and better monitoring methods. Currently, the methods used for early diagnosis and prognosis of tumors involving Western blotting, enzyme-linked immunosorbent assays, and flow cytometry all face technical barriers due to their limited sample sizes, low sensitivity, or the need for specialized high-end equipment [2]; therefore, electrochemical biosensors are a new direction for researchers in the detection of tumor cells.

An electrochemical biosensor is a device that converts an interaction signal between a biometric element and a recognition target into a detectable electrical signal [3]. Over the past decade, many papers related to electrochemical sensors have been published, proving that the development of electrochemical sensors is still a popular research area. In general, electrochemical biosensors have the advantages of excellent specificity, high sensitivity, simple equipment, and low cost, having attracted extensive attention in various fields, including for tumor diagnosis and treatment. For example, an electrochemical cell sensing technique combined with a biochip [4] was reported by Suhito et al. for the detection of human glioblastoma cells and for drug evaluation, which showed satisfactory results owing to the excellent performance of the biochip [5]. Living cells have been employed in biosensors as biometric elements since the early 1970s [6]. Cells are excellent bioreceptors owing to their flexibility in determining sensing strategies, which are cheaper than purified enzymes and antibodies, making the manufacture of sensors simpler and more efficient. Tumor cells can express multiple molecules at different levels, which facilitate quantitative analysis of analytes, saving time and costs. Compared with molecular-based biosensors, cell-based biosensors provide convenience in designing functional strategies.

In recent years, more publications related to cell-based biosensors have been published and many reviews have been reported [7,8], although few reviews related to electrochemical sensors based on tumor cells have been reported. In this review, we present the results of various electrochemical cytosensors for cancer detection that have been reported in the literature over the past five years. Finally, some recent trends and prospects for cell-based biosensors are discussed.

**Figure 1 ijms-22-08184-f001:**
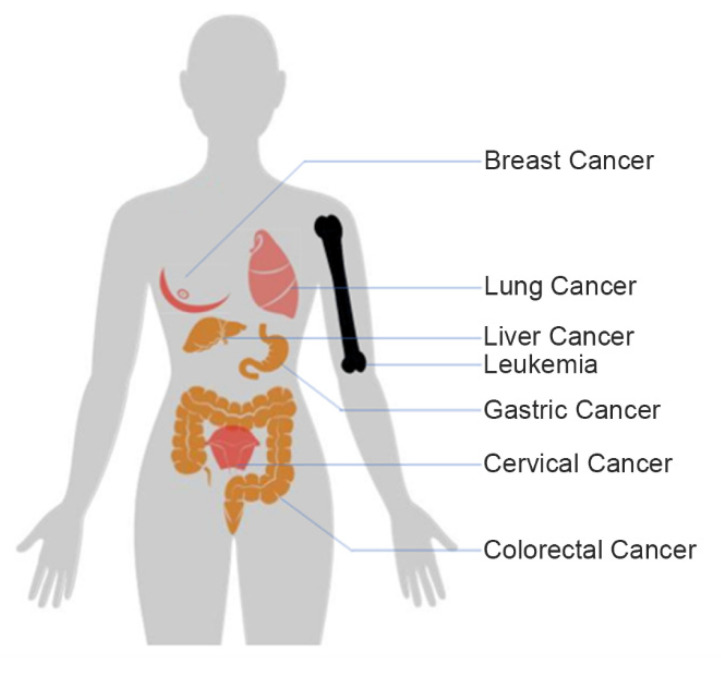
Distribution of different kinds of cancers in the human body. Figure was drawn according to reference [9].

## 2. Electrochemical Detection of Markers on Tumor Cells

As a subclass of electrochemical biosensors, electrochemical cytosensors consist of a biometric element, an electrochemical sensor, and cells. Some also contain probes to amplify signals. Through electrochemical technology, the interactions between biometric elements (antibodies, aptamers, lectins, etc.) and target cells are converted into electrical signals, which are converted into usable information through a transducer and signal processor [10]. Electrochemical cell sensors used for detection of markers on tumor cells usually use the sandwich model, as exhibited in Figure 2. In addition, some special electrochemical cell sensors are used to detect cancer cells [11]. We reviewed some of the electrochemical cell sensors used to detect the markers on tumor cells over the last five years.

### 2.1. Breast Cancer

Breast cancer has surpassed lung cancer and become the most common cancer, with 2.3 million new cases occurring in 2020 and accounting for 11.7 percent of all new cancer cases [1]. Although the incidence rate of breast cancer is increasing, the global mortality rate of breast cancer is gradually decreasing owing to the popularization of new treatment strategies and methods; however, breast cancer is still one of the leading causes of worldwide death among women [12] and it is important to develop new alternative methods to diagnose breast cancer.

Transmembrane glycoprotein mucin 1 (MUC1), nucleoli [13], and epidermal growth factor receptor (EGFR) are frequently used markers in the electrochemical detection of breast cancer cells, which have been proven to be highly overexpressed on the surfaces of breast cancer cells [14]. Ou et al. constructed a sandwich-cell-based sensor using two aptamers (AS1411 and MUC1) linked with a metal–organic framework (PCN-224) and tetrahedral DNA nanostructures (TDNs) to detect breast cancer cells [15]. TDNs linked with dual aptamers immobilized on the gold electrode (GE) increased the density and orientation of the surface nanoprobe. A probe modified with PCN-224 as the skeleton was used to amplify the electrochemical signal. The sensor exhibited a wide linear range (20–1 × 10^7^ cells/mL) and a low detection limit (6 cells/mL), as detected by differential impulse voltammetry (DPV). Song and his team synthesized a novel nanocomposite, which was used for construction of an electrochemical-cell-based biosensor and detection of EGFR overexpressed in MCF-7 cells [16]. A new bimetallic CuCo Prussian blue analogue (CuCo PBA) loaded with carbon dots (CDs) (CD@CuCoPBA) presented a nanocube shape and retained its nanostructure and physicochemical properties, showing excellent electrochemical activity and high stability. With the biocompatible nanocomposite composed of multiwalled carbon nanotubes and polyglutamate, Yazdanparast et al. developed an aptamer-based sandwich method for sensitive and selective detection of human breast cancer cells and MUC1 biomarkers [17]. MCF-7 cells were captured by aptamers fixed on the electrode, then the aptamers labeled with silver nanoparticles were used for secondary cell recognition to increase the selectivity and amplify the signal. Under optimal conditions, the cell-based biosensor responded to MCF-7 cells within a concentration range of 1.0 × 10^2^–1.0 × 10^7^ cells/mL, with a detection limit of 25 cells/mL. Yang et al. prepared a multiaptamer DNA tetrahedron nanostructure and applied it and the MUC1 aptamer to the cell-based biosensor. After a series of optimization processes, they finally obtained a high-quality sensor for MCF-7 cells, with a detection range of 50–1.0 × 10^6^ cells/mL and a detection limit of 5 cells/mL [18]. In order to prevent nonspecific binding and improve specificity, Liu et al. developed an antifouling electrochemical-cell-based biosensor to detect MCF-7 cells in a complex biological environment [19]. In this work, polyaniline membrane was electrodeposited on an electrode to support the MUC1 aptamer and an investigator-designed antifouling branched peptide (Figure 3). The reported cell-based biosensor had an excellent linear range (50–1.0 × 10^6^ cells/mL) and detection limits (20 cells/mL) for MCF-7 cells.

The expression level of CD44 on the cell surface is closely related to the metastatic potential of cancer cells [20]. Hyaluronic acid (HA) is the main ligand of CD44, having a high affinity [21]; therefore, Zhou et al. [22] developed a novel unlabeled electrochemical impedance spectroscopy (EIS) sensor by successfully utilizing HA combined with gold nanoparticles (GNPs). The sensor was based on molecular recognition by HA, which could capture CD44 overexpressed cancer cells using nanocomposite-modified electrodes to increase electron transfer resistance. The changes of resistance can be used as a signal to evaluate the number of cancer cells and the expression level of CD44. Another electrochemical aptamer sensor takes advantage of the high affinity and specificity of AS1411 aptamer to nucleolin [23]. In their study, Farzin and his colleagues synthesized an ionicliquid–hydroxyapatite nanorod (IL/HApNR)-AuNP nanocomposite in order to design an amplifier nanoplatform and fix the aptamer. When the MCF-7 target cell was present, the signal probe was replaced and released from the electrode surface, which led to a decrease of the current in proportion to the increasing concentration of the cancer cells in the range of 10 to 1.0 × 10^6^ cells/mL, with a detection limit of 8 ± 2 cells/mL.

It is well known that there are different subtypes of breast cancer with different tumor-causing and metastatic abilities [24]. Different subtypes of breast cancer cells have different gene expression levels and different intracellular and extracellular molecules, which likes a fingerprint and can be used to identify different cell subtypes. The lectins extracted from plants or animals are common glycoprotein or protein, which can bind to specific glycans on the surfaces of cell membranes [25]. Zanghelini and his coworkers developed a lectin-based biosensor that enabled them to distinguish normal human skin fibroblast (NHSF) cell lines, less aggressive MCF-7, and more aggressive T47D cancer cell lines via outer-membrane carbohydrate profiles [26]. The senor achieved a minimum detection limit of 10 cells/mL and a linear range of 10–1.0 × 10^6^ cells/mL. Han et al. designed an automated DNA assembly reaction and used it for the simultaneous identification of dual therapeutic targets using electrochemical techniques [27]. Using MDA-MB-231 breast cancer cells as a model, the capture probe based on quantum dots was used to recognize the surface biomarkers EGFR and ICAM-1, which induced a chain-like chain replacement reaction. The electrochemical results showed that the method was specific and sensitive to target sites. Inspired by the morphology and shape characteristics of breast cancer cells, Wang et al. developed a three-dimensional biological interface in a graphene-based electrical impedance sensor for diagnosis of single-cell-resolution metastatic cancer [28]. The graphene biological interface simulated the morphology and body shape characteristics of cancer cells, while the contact area between the cell and the graphene electrode was significantly increased, allowing more comprehensive and thorough single-cell signals in three-dimensional space. The electrical signals were about twice as strong as those collected at the two-dimensional gold interface.

At present, the electrochemical detection of breast cancer cells is mostly based on the interactions between proteins expressed at different levels between cancer and normal cells and antibodies, aptamers, and small-molecule-targeting substances fixed on the electrode. Unmixed electrochemical technology is mature but it is difficult to achieve breakthroughs with this technology within a short timeframe. Most researchers rely on novel modification materials and capture probes to improve the performance of electrochemical-cell-based biosensors (Table 1). As of now, electrochemical-cell-based biosensors that intersect with other disciplines or detection technologies are still in the minority.

### 2.2. Lung Cancer

Although lung cancer (11.4%) was surpassed by breast cancer (11.7%) in terms of total new cancer cases, it remains the leading cause of cancer death, with approximately 1.8 million deaths (18%) in 2020 [1]. Because the early symptoms of lung cancer are not obvious, methods for early detection and treatment are in demand [29]. Recently, Bolat et al. used polydopamine nanoparticles for the first time for the unlabeled electrochemical determination of A549 lung cancer cells [30]. Using the self-polymerizing ability of controlled dopamine, Bolat and his colleagues constructed an electrochemical probe on the surface of a graphite electrode. The developed sensor displayed good biocompatibility, with a detection range of 1.0 × 10^2^–1.0 × 10^5^ cells/mL and a minimum detection of 25 cells/mL for A549 cells under suitable conditions. The epithelium–mesenchymal transformation (EMT) can promote the migration of tumor cells [31], which is of great significance for cancer detection. Our lab demonstrated an electrochemical biosensor for the detection of EMT based on E-cadherin, an important marker of EMT [32]. We used E-cadherin antibody quantum dots as a multifunctional signal probe and modified the sensing platform with carbon nanotubes and gold nanoparticles. In addition, the biosensor we developed can detect tumor cells and distinguish between in situ and circulating tumor cells, providing an excellent linear range (75–5.5 × 10^4^ cells/mL) and detection limit (75 cells/mL) for A549 cells. Transferrin receptors, which are abundantly expressed in tumor cells, help to absorb iron and participate in cellular life activities [33]. Based on this, de Almeida et al. created an electrochemical biosensor that indirectly detected cancer cells [34]. Polyclonal antibody of transferrin (anti-TF) was immobilized on the electrode to indirectly detect cancer cells by detecting the binding of anti-TF to different types of transferrin through the Fe adhesion cycle between the transferrin and its receptor. The sensor exhibited a detection limit of 10^2^ cells/mL. Wang et al. proposed a sensing platform to detect A549 cells using the principle of photochemistry [35]. In this study, an indium tin oxide electrode was modified with iron phthalocyanine and Ag–ZnIn_2_S_4_ quantum dots in order to generate electrical signals under near-infrared light irradiation. The hyaluronic acid was used to capture A549 cells, resulting in the blockage of electron transport and the reduction of electrical signals (Figure 4). The sensing platform had an excellent linear range (2.0 × 10^2^–4.5 × 10^6^ cells/mL) and detection limit (15 cells/mL) for A549 cells. In addition, the cytosensors created by Zhang et al. (marker: carcinoembryonic antigen) [36], Ma et al. (marker: sialic acid) [37] and Zhang et al. (marker: nucleoli) [38] were all excellent products for the detection of marker of A549 cells (Table 2). With the development of electrochemical detection of breast cancer cells, most researchers have sought breakthroughs for the electrode materials and probes. Some researchers have combined electrochemical detection with other disciplines or detection technologies, such as the above photoelectrochemical detection and EMT detection approaches.

### 2.3. Cervical Cancer

Cervical cancer is the second most common cancer among women worldwide. The age range of high incidence of carcinoma in situ is 30 to 35 years old, while that of invasive carcinoma is 45 to 55 years old. In recent years, the incidence of the disease has tended to be younger [39]. Rapid and accurate determination of the tumor development period is key for successful treatment of cervical cancer. In order to prevent false positives and interference from other substances, Fan et al. first reported an antifouling photoelectrochemical (PEC) cytosensor involving aptamer AS1411 and zwitterionic peptide [40]. TiO_2_ and ZnIn_2_S_4_ nanoparticles were decorated on the ITO electrode in turn, then the aptamer and amphoteric peptide were immobilized on the modified electrode. The sensitivity of the developed sensor was very high, with a good linear relationship in the range of 1.0 × 10^2^–1.0 × 10^6^ cells/mL, while the detection limit was 34 cells/mL for HeLa cells. Zhou et al. developed a sandwich electrochemical sensor with tyrosine signal amplification to detect HeLa cells [41]. They immobilized the aptamer AS1411 on a gold electrode to capture HeLa cells and used a platinum–horseradish peroxidase–aptamer AS1411 complex as a probe to form a sandwich structure. The sensor used carbon diamine coupling reaction to amplify the signal though tyramine-functionalized infinite coordinate polymers (ICPs@Tyr) as a biological conjugate. In this system, the functionalized platinum nanoparticles on the probe improved the catalytic performance of horseradish peroxidase, resulting in the continuous deposition of the labeled-signal ICPs@Tyr on the cells and improving the detection of rare tumor cells, with linear concentrations ranging from 2 to 2.0 × 10^4^ cells/mL and detection limits as low as 2 cells/mL. Dutta et al. designed a multifunctional nanocomposite material for electrochemical cancer detection and targeted therapy [42]. In this work, graphene quantum dots (GQDs) and Fe_3_O_4_ nanocomposites combined with lectin protein were deposited on a Pt electrode to form a novel multifunctional nanocomposite for detection of HeLa cells. The dynamic linear range was 5.0 × 10^2^–1.0 × 10^5^ cells/mL, while the detection limit was 273 cells/mL. Using water-dispersible 2D graphite-like carbon nitride nanosheet–silver iodide nanocomposites and anti-CEM/PTK7 aptamers as biometric elements, Mazhabi and his team designed a novel, label-free PEC aptamer cell-based biosensor with a response range of 10 to 1.0 × 10^6^ cells/mL [43]. The designed cell-based biosensor had low overpotential and good reproducibility, stability, and specificity, and was used for the determination of HeLa cells at concentrations as low as 5 cells/mL.

The electrochemical detection of cervical cancer cells usually involves HeLa cells as the model cells (Table 3). At present, only the AS1411 aptamer can be used to capture HeLa cells [9]; therefore, new capture probes and new electrode modification materials need to be developed to improve the sensor performance.

### 2.4. Liver Cancer

Liver cancer is one of the most common cancers [45], which is particularly difficult to detect and easily ignored in the early stages of symptoms. By the time a person becomes aware of the illness, it is usually too late to be treated successfully; therefore, the development of a novel simple and accurate detection method for liver cancer would be very significant for such patients (Table 4). Li and colleagues reported on a universal method for signaling on the cell surfaces to detect cancer cells [46]. The cancer cells were linked to DNA-bridge-complex-templated silver nanoclusters (DNA bridge-AGNCs) by opening cell membrane protein disulfide bonds with a thiol–maleimide conjugation. Li et al. [4] used 4-sulfocalix arene hydrate and antibodies (anti-MUC1) fixed to modify electrodes to capture HepG2 cells and DNA bridge-AGNCs to amplify the signal. The electrochemical test results showed a wide linear range of 50–2.0 × 10^6^ cells/mL and a detection limit as low as 15 cells/mL. In another study, Li and colleagues demonstrated an electrochemical sensor based on folic acid (FA) and octadecylamine (OA)-functionalized graphene aerogel microspheres (FA-GAM-OA) for detection of HepG2 cells [47]. The FA-GAM-OA synthesized using the Pickering emulsion method differs from ordinary GO in that it consists of smaller graphene sheets. It has a larger surface area, which ensures full exposure to more folic acid heads and improves the sensitivity and selectivity. The electrochemical sensor showed commendable analytical performance in the detection of HCC cells, with a linear range of 5–10^5^ cells/mL and a low detection limit of 5 cells/mL. This method has already been used to detect cancer cells in whole blood. Zheng et al. proposed a strategy for the electrochemically sensitive detection of HepG2 cells [48]. They modified the EpCAM aptamer binding to cells and added polyadenine at the 3′-OH terminus of the aptamers. The aptamer with the polyadenine tail was adsorbed to the surface of the gold electrode to obtain an electrical signal (Figure 5). Using this strategy, responses can be obtained in the linear range of 10–10^3^ cells/mL, with a detection limit as low as 3 cells/mL.

In recent years, DNA tetrahedral nanostructures have been widely used in electrochemical sensors. The DNA nanotetrahedron is fixed to the gold electrode, creating a well-designed platform to capture cells more specifically and efficiently. Sun et al. developed a label-free competitive electrochemical-cell-based biosensor with a sandwich structure [49]. The DNA nanotetrahedron–TLS11a aptamer was immobilized on a silk-printed gold electrode to capture cells, and hybrid nanoprobe-labeled Pd-Pt nanospheres with complementary DNA, heme/G-quad dioxygen–ribonuclase, and horseradish peroxidase were used to amplify the signal. When HepG2 cells are present, they can compete with the nanoprobes and bind to the TLS11a aptamer, causing the probes to be released from the disposable screen-printed gold electrode and the electrochemical signal to change. Under suitable conditions, the electrochemical analysis showed a linear correlation in the range of 10–10^6^ cell/mL, with a minimum detection limit of 5 cells/mL. A few months later, Sun et al. published another paper in which they modified the signal probe to introduce a rolling loop amplification (RCA)-directed deoxyribozyme strategy [50]. The signal probe consisted of a carboxyluciferin (FAM)-functionalized TlS11a aptamer, DNA primer chain, Pt nanoparticles, and horseradish peroxidase. G-quadruplex/hemin DNAzyme was generated to amplify the signal by using primer DNA and a circular probe. After modification, the linear range of electrochemical detection was unchanged, although the sensitivity was greatly improved and the detection limit was reduced to 3 cells/mL. Around the same time, Chen published a paper [51]. The cell-based biosensor (composed of DNA nanotetrahedrons and composite functional probes) proposed by Chen offered a wide detection range, with a lower limit of 5 cells/mL. Cancer stem cells (CSCs) are thought to be responsible for cancer recurrence and resistance to chemotherapy [52]. Eissa’s group reported on a label-free impedance cell-based biosensor using antibodies specific to four established tumor stem cell surface biomarkers (CD44, CD90, CD133/2, and OV-6) [53]. Cysteamine–phenylene isothiocyanate was used to attach the antibody to a gold electrode to capture the CSCs. The sensor’s linear range was from 10 to 10^4^ cells/mL, with a detection limit of 1 cell/mL.

In the construction of electrochemical-cell-based biosensors that detect cancer cells, nucleic acid technologies such as the aforementioned RCA and DNA walker methods are often used to amplify signals. These nucleic acid technologies have strong ability to amplify signals and have advantages in detecting trace cells and low-expression proteins. At present, research in this area is scarce.

**Table 4 ijms-22-08184-t004:** Electrochemical cytosensors used for liver cancer cell detection.

Cell Types	Cancer Markers	Modification Material	Linearity Range (Cells/mL)	Limit of Detection (Cells/mL)	Reference
HepG2	MUC1	pSC_4_	50–2 × 10^6^	15	[46]
HepG2	FA receptors	FA-GAM-OA	5–1 × 10^5^	5	[47]
HepG2	EpCAM	—	10–1 × 10^3^	3	[48]
HepG2	Membrane surface	DNA nanotetrahedron	10–1 × 10^6^	5	[49]
HepG2	EpCAM	PAMAM	1 × 10^4^–1 × 10^6^	2.1 × 10^3^	[54]
HepG2	Membrane surface	ZnO@Au-Pd NPs	1 × 10^2^–1 × 10^7^	10	[55]
CSCs	CD44, CD90, CD133/2, and OV-6	PDITC	10–1 × 10^4^	1	[53]

4-Sulfocalix arene hydrate, pSC_4_; polyamidoamine dendrimer, PAMAM; cysteamine–phenylene isothiocyanate, PDITC.

### 2.5. Leukemia Cells

Leukemia is a malignant clonal disease of hematopoietic stem cells [56]. Due to mechanisms such as uncontrolled proliferation, differentiation disorder, and blocked apoptosis, clonal leukemia cells proliferate and accumulate in bone marrow and other hematopoietic tissues, infiltrate other nonhematopoietic tissues and organs, and inhibit normal hematopoietic function. According to different prognostic indicators, leukemia patients can be divided into different prognostic levels for different treatment intensities; therefore, it is important to complete the comprehensive examinations required for various prognostic stratification processes and to develop a personalized treatment plan. Sugawara et al. found and designed an electronic transfer peptide, YYYYC [57], which had excellent electroactivity and the functionality of two hydrate-mimicking peptides [58]. They then performed an electrochemical analysis of the changes in current based on the competition between YYYYC and cancer cells for SBA. In the subsequent study [59], Sugawara et al. cross-linked YYYYC with myelopeptide-4 (MP-4: FrPrimTP), a peptide with recognition function derived from bone marrow. The YYYYC and MP-4 were fixed on the electrode using collagen to perform electrochemical impedance spectroscopy. Although the electrochemical response range was reduced, the detection limit was as low as 8 cells/mL.

Li et al. published a photoelectric chemobiosensor based on hypotoxic ternary mercaptopropionic acid (MPA)-capped AgInS2 nanoparticles (NPs) [60]. Under the excitation of red light, AgInS2 nanoparticles showed extremely high photoelectric conversion efficiency and generated a strong light current, which improved the detection sensitivity. The current intensity decreased when the sgc8c aptamer captured CCRF-CEM cells. The sensor had excellent specificity, with a minimum detection limit of 16 cells/mL. In another study, Wang and colleagues proposed a sandwich electrochemical-cell-based biosensor of K562 cells based on quantum-dot-functionalized microspheres [28]. In this study, they synthesized polystyrene microsphere–CdS QDs–Con A conjugates as a signal probe and used graphene oxide–polyaniline–glutaraldehyde–Con A to modify electrode. The biosensor was quite sensitive, with a linear range of 10–1.0 × 10^7^ cells/mL and a minimum detection limit of 3 cells/mL.

Khoshroo et al. reported on a sandwich-type electrochemical aptamer cell-based biosensor for detecting CCRF-CEM cells [61]. The sensor used a copper sulfide–graphene nanocomposite as the signal material and used a gold–graphene nanocomposite to modify the electrode to capture cells through the aptamer. The copper sulfide–graphene nanocomposite material increased the sensitivity of the biosensor. Based on the principle of plasmon-enhanced electrochemistry, Wang et al. developed an unlabeled electrochemical-cell-based biosensor for detection of CCRF-CEM cells [62]. Wang immobilized sgc8c aptamers on electrodes dripped with gold nanostars. Ascorbic acid was oxidized by the electrode surface, which amplified the signal due to the excitation of compacting surface plasmon resonance under light. As the aptamer trapped more cells and the surface of the electrode was covered, the electrical signal dropped (Figure 6). The reported biosensor’s response to the current was over the linear range of 5–10^5^ cells/mL, with a minimum detectable concentration of 5 cells/mL.

Leukemia has many model cells and biomarkers [63]. The construction of electrochemical-cell-based biosensors for detection of leukemia cells provides many options for capturing probes (Table 5).

### 2.6. Gastric Cancer and Colorectal Cancer

Gastric cancer is a malignant disease with high morbidity and mortality [1]. By virtue of the interaction between the boronic acid and cell-surface-locating carbohydrates [68], Dervisevic et al. designed an electrochemical cytosensor based on boric-acid-functionalized polythiophene to detect gastric cancer cells, which was prepared via electropolymerization of 3-thienyl boronic acid and thiophene coated on a graphite electrode [69]. After ten minutes of cell incubation, the cytosensor showed extremely high analytical performance and selectivity for AGS cells, with an analytical range of 10–1 × 10^6^ cells/mL and a minimum detection limit of 10 cells/mL. In the same year, the team published another paper for the detection of gastric cancer cells. In this study, Dervisevic and his team refined their sensor to develop folic-acid- and boric-acid-based cytosensors for cancer detection and performance comparison [70]. The gold electrodes were modified with cysteamine and immobilized with a ferrocene-cored polyamine diamine dendrimer. Then, the electrodes were modified with either folic acid or boric acid. The detection limits for the FA-based electrode and the BA-based electrode were 20 and 28 cells/mL, respectively. Tabrizi et al. developed a sandwich-type electrochemical-cell-based biosensor to detect AGS cells [71]. In this study, Tabrizi and co-workers used aptamer–Au@Ag nanoparticles as a signal probe to detect ASG cells for the first time. The primary aptamers were immobilized on electrodes modified by MWCNT-Au nanocomposites to capture AGS cells. The secondary aptamer of the signal probe combines with cells and electrode to form a sandwich structure. Electrochemical detection was carried out in the hydrogen peroxide flow channel. The sensor was also applied for the determination of AGS cancer cells in human serum samples with good selectivity and stability.

Colorectal cancer is the third most common cancer worldwide (accounting for 10.0% of new cancer cases) and the second most common cause of cancer death worldwide (9.4% of cancer deaths) [1]. Early diagnosis of colorectal cancer is important for successful treatment and can reduce mortality. Duan et al. proposed a new electrochemical-cell-based biosensor to detect CT26 colorectal cancer cells based on a series of nanohybrids of a Cr-based metal–organic framework (Cr-MOF) and cobalt phthalocyanine (CoPc) nanoparticles (Cr-MOF@CoPc) [72], which had excellent electrochemical properties and strong fluorescence. The demonstrated sensor was capable of sensing analysis in the range of 50–1 × 10^7^ cells/mL, with a minimum detection limit of 8 cells/mL. In a separate study, Akbal and his colleagues published an electrochemical-cell-based biosensor based on folic-acid-doped Prussian blue nanoparticles (FA-PB NPs) [73]. FA increased the stability and biocompatibility of FA-PB NPs. DLD-1 cells were fixed on a FA-PB NP-modified electrode based on the interaction between FA and the overexpressed FA receptor on the cell surface. The cell-based biosensor had an excellent electrochemical sensing signal in the range of 5.0 × 10^2^–1.0 × 10^5^ cells/mL, with a minimum detection limit of 48 cells/mL. Using functionalized fibrous nanosilica (KCC-1) and folic acid, Soleymani et al. demonstrated a specific electrochemical detection method for HT29 cells [74]. KCC-1 was synthesized using the hydrothermal method and functionalized KCC-1-NH_2_ was synthesized via a reaction with APTES. The product of the previous step was cross-linked with FA through EDC/NHS to form KCC-1-NH_2_-FA. The interaction between FA and FA receptors overexpressed on the surface of cancer cells was also used to recognize cells.

The probes used to capture these two types of cancer cells are mostly small-molecule-targeting substances (Table 6); further study is required of the related biomarkers on cell membranes and for improvement of the probes.

## 3. Conclusions

Electrochemical-cell-based biosensors have emerged as some of the most promising alternatives to traditional cancer detection techniques. In this review, electrochemical-cell-based biosensors used for tumor cells detection in the past five years were assessed. It makes sense that researchers are constantly using new modifiers and new targets to improve the performance of sensors. In addition, in recent years, more studies have been conducted on the combination of a variety of monitoring technologies to prepare electrochemical cells based on biosensors, using optical and electrochemical technology, biochip, microfluidic technology, and DNA walker techniques. The combination of these new technologies provides advantages over traditional technologies. Unfortunately, the existing electrochemical cell sensing technology has not been widely used in clinical practice. This may be due to the difficulty of sensing detection in vivo and the lack of specificity for the captured cancer cells. Current electrochemical cell sensing strategies lack the ability to detect intracellular protein markers. In addition, electrochemical-cell-based biosensors for the detection of living cells and single cell need to be developed; therefore, the application of electrochemical-cell-based biosensors in the detection of tumor cells needs to be further explored.

## Figures and Tables

**Figure 2 ijms-22-08184-f002:**
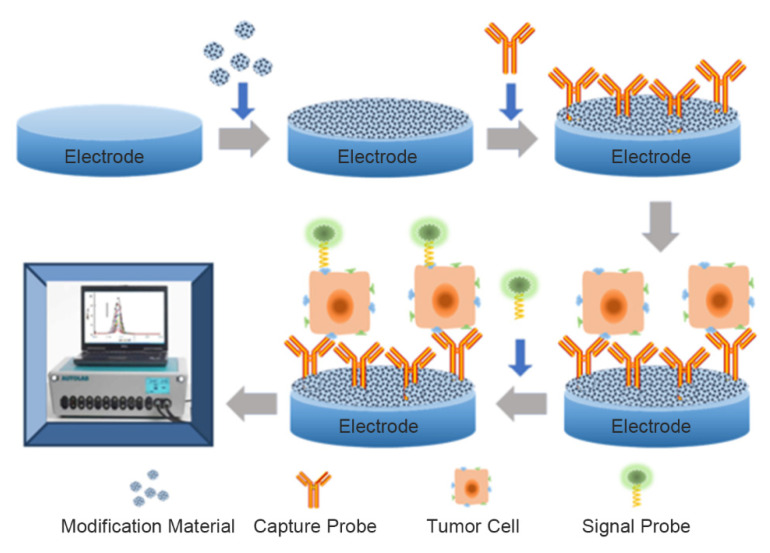
Composition of electrochemical cytosensors.

**Figure 3 ijms-22-08184-f003:**
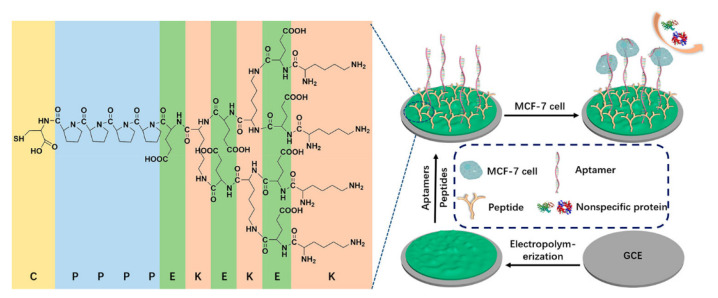
Schematic diagram of antifouling electrochemical-cell-based biosensor. The structure of the synthetic branched peptide is shown on the left. Reproduced with permission from [19], American Chemical Society, 2019.

**Figure 4 ijms-22-08184-f004:**
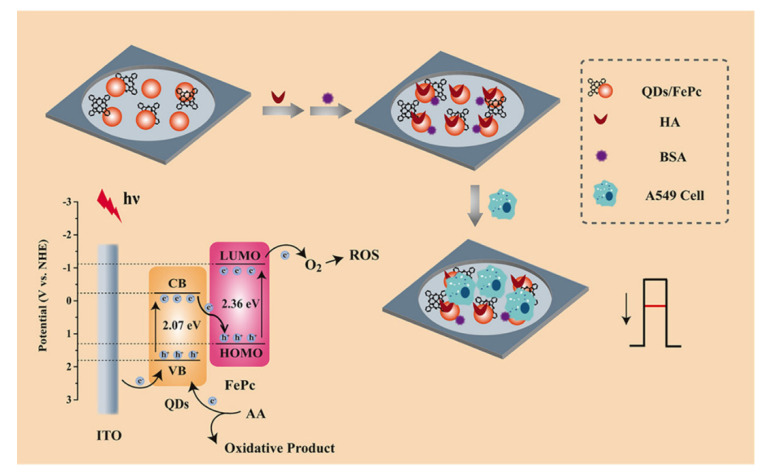
Schematic diagram of the sensing platform proposed by Wang. Reproduced with permission from [35], American Chemical Society, 2020.

**Figure 5 ijms-22-08184-f005:**
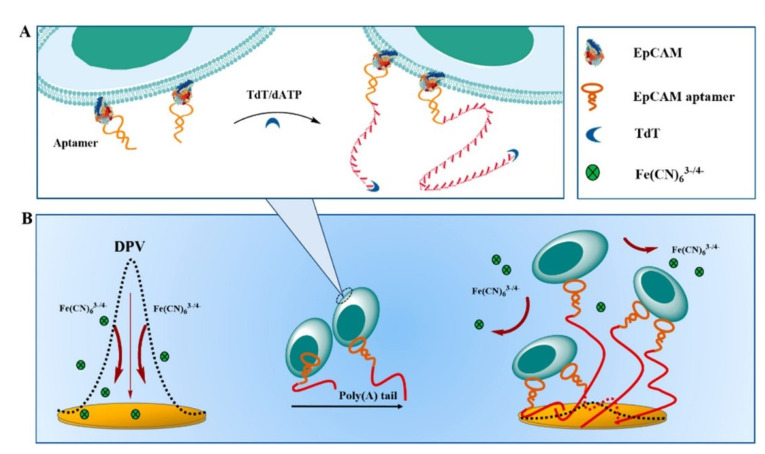
(**A**) The addition of polyadenine at the 3′-OH terminus of the aptamers. (**B**) Schematic diagram of the strategy used for electrochemically detection. Reproduced with permission from [48], American Chemical Society, 2020.

**Figure 6 ijms-22-08184-f006:**
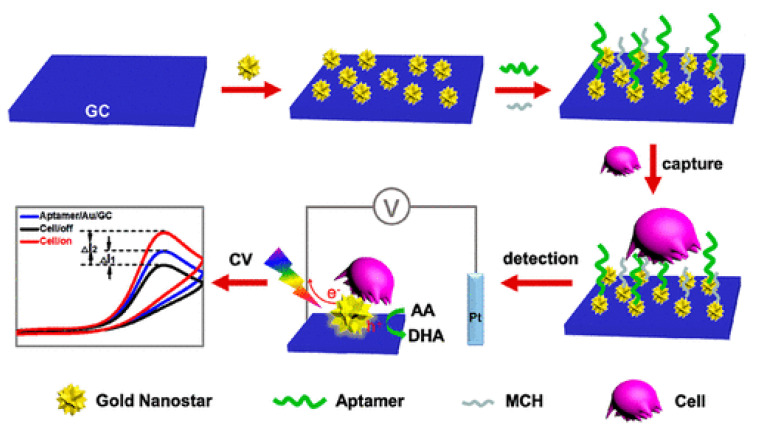
Schematic diagram of the cell-based biosensor developed by Wang. Reproduced with permission from [62], American Chemical Society, 2019.

**Table 1 ijms-22-08184-t001:** Electrochemical cytosensors for breast cancer cell detection.

Cell Types	Cancer Markers	Modification Material	Linearity Range (Cells/mL)	Limit of Detection (Cells/mL)	Reference
MCF7	MUC1 and Nucleoli	TDNs	20–1 × 10^7^	6	[15]
MCF7	EGFR	CD@CuCoPBA	5 × 10^2^–1 × 10^5^	80	[16]
MCF7	MUC1	Apt-DTNs	50–1 × 10^6^	5	[18]
MCF7	MUC1	PANI Films	50–1 × 10^6^	20	[19]
MCF7	Nucleoli	IL/HAp-Au NPs	10–1 × 10^6^	8 ± 2	[23]
MCF7	MUC1	MWCNT-PGA	1 × 10^2^–1 × 10^7^	25	[17]
MDA-MB-231	CD44	GNPs	2 × 10^2^–3 × 10^5^	128	[22]
T47D	Carbohydrate profile	TiO_2_-MN	10–1 × 10^6^	10	[26]

Polyaniline, PANI; multiwalled carbon nanotubes, MWCNT; polyglutamate, PGA; TiO_2_ membrane nanostructures, TiO_2_-MN.

**Table 2 ijms-22-08184-t002:** Electrochemical cytosensors used for lung cancer cell detection.

Cell Types	Cancer Markers	Modification Material	Linearity Range (Cells/mL)	Limit of Detection (Cells/mL)	Reference
A549	—	PDA NPs	1 × 10^2^–1 × 10^5^	25	[30]
A549	E-cadherin	CNT-Au NPs	75–5.5 × 10^4^	75	[32]
A549	CD44	FePc/Cys AZIS QDs	2 × 10^2^–4.5 × 10^6^	15	[35]
A549	Carcinoembryonic antigen	CNS@AuNP	42–4.2 × 10^6^	14	[36]
A549	Sialic acid	CS-Au/hPPy	10–1 × 10^7^	2	[37]
A549	Nucleoli	PGO Film	—	10	[38]

Polydopamine nanoparticles, PDA NPs; carbon nanotubes and gold nanoparticles, CNT-Au NPs; iron phthalocyanine and cysteine-modified Ag–ZnIn_2_S_4_ quantum dots, FePc/Cys AZIS QDs; monodisperse colloidal carbon nanospheres, CNSs; hollow horn-like PPy, hPPy; chitosan, CS; porous graphene oxide, PGO.

**Table 3 ijms-22-08184-t003:** Electrochemical used cytosensors for cervical cancer cell detection.

Cell Types	Cancer Markers	Modification Material	Linearity Range (Cells/mL)	Limit of Detection (Cells/mL)	Reference
HeLa	Nucleoli	ITO/TiO_2_/ZnIn_2_S_4_	1×10^2^–1 × 10^6^	34	[40]
HeLa	Nucleoli	ICPs@Tyr	2–2 × 10^4^	2	[41]
HeLa	Carbohydrate profile	GQDs-Con A@Fe_3_O_4_	5 × 10^2^–1 × 10^5^	274	[42]
HeLa	CEM/PTK7	WDg-C_3_N_4_-AgI	10–1 × 10^6^	5	[43]
HeLa	Folate receptors	Au-NaYF_4_:Yb.Er	4.25 × 10^2^–4.25 × 10^5^	326	[44]

Water-dispersible 2D graphite-like carbon nitride nanosheet–silver iodide nanocomposites, WDg-C_3_N_4_-AgI); upconversion nanoparticles of yttrium tetrafluoride doped with erbium and ytterbium, NaYF_4_:Yb.Er.

**Table 5 ijms-22-08184-t005:** Electrochemical cytosensors used for leukemia cell detection.

Cell Types	Cancer Markers	Modification Material	Linearity Range (Cells/mL)	Limit of Detection (Cells/mL)	Reference
K562	ASGP receptors	—	1 × 10^2^–5 × 10^3^	—	[58]
K562	CD44	Collagen and Peptide	27–2 × 10^3^	8	[59]
K562	P-glycoprotein	PET and MWNT	1.5 × 10^2^–1.5 × 10^7^	10	[64]
K562	Carbohydrate profile	GO-PANI-GA-Con A	10–1 × 10^7^	3	[28]
K562	Carbohydrate profile	—	1 × 10^2^–1 × 10^7^	79	[65]
HL-60	—	C-MWNT	1 × 10^2^–1 × 10^7^	35	[66]
CCRF-CEM	PTK-7	AgInS_2_	1.5 × 10^2^–3 × 10^5^	16	[60]
CCRF-CEM	PTK-7	Au-GR NPs	50–1 × 10^6^	18	[61]
CCRF-CEM	PTK-7	AuNSs	5–1 × 10^5^	5	[62]
CCRF-CEM	PTK-7	MWNT-Pd NPs/PTCA	10–5 × 10^5^	8	[67]

Polyethylene terephthalate, PET; graphene oxide, GO; glutaraldehyde, GA; carboxylic-group-functionalized multiwalled carbon nanotubes, C-MWNT; graphene, GR; gold nanostars, Au NSs; perylene tetracarboxylic acid, PTCA.

**Table 6 ijms-22-08184-t006:** Electrochemical cytosensors used for gastric cancer cell and colorectal cancer cell detection.

Cell Types	Cancer Markers	Modification Material	Linearity Range (Cells/mL)	Limit of Detection (Cells/mL)	Reference
AGS	Sialic acid	p(TBA_0.5_Th_0.5_)	10–1 × 10^6^	10	[69]
AGS	Folate receptors and Sialic acid	Au/Fc-PAMAM(G2)/FA and Au/Fc-PAMAM(G2)/BA	1 × 10^2^–1 × 10^6^	20/28	[70]
AGS	Carbohydrate profile	MWCNT-Au NPs	10–5 × 10^5^	6	[71]
BGC-823	Mannosyl groups and Sialic acid	TGA-Au@BSA microspheres-APBA	1 × 10^2^–1 × 10^6^	40	[75]
BGC-823	Mannosyl groups and Sialic acid	Au@BSA microspheres-Con A	5 × 10^2^–1 × 10^6^	120	[76]
CT26	—	Cr-MOF@CoPc	50–1 × 10^7^	36/8	[72]
DLD-1	FA receptors	FA-PB NPs	5 × 10^2^–1 × 10^5^	48	[73]
HT29	FA receptors	KCC-1-NH_2_-FA	50–1.2 × 10^4^	50	[74]

Electropolymerization of 3-thienyl boronic acid and thiophen, p(TBA0.5Th0.5); second-generation polyamidiamine dendrimers, PAMAM (G2); ferrocene, Fc; thioglycolic acid, TGA; bovine serum albumin, BSA; 3-aminophenylboronic acid, APBA.

## Data Availability

Not applicable.

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
