# Peer review of "Electrochemical Sensors for Detection of Markers on Tumor Cells"

_ijms, 2021, doi:10.3390/ijms22158184_

Round 1
Reviewer 1 Report
The review describes the results of various electrochemical cytosensors for cancer detection that have been reported in the literature over the past five years. The work is a significant contribution to the field. It is well structured. The introduction is presented the most threatening disease in the world, cancer, because of its high morbidity and mortality rate. Also, several methods used for early diagnosis and prognosis of tumor are named and compared with electrochemical sensors advantages.
After the state of the art in the domain, the electrochemical detection of markers on tumor cells is illustrated for seven types of cancer: breast, lung, cervical, liver, leukemia, gastric and colorectal.
The schematic diagram of the electrochemically sensing platform proposed in the literature is shown in the figures and some electrochemical cytosensors for cancer cell detection with construction material and electroanalytical characteristics are enumerated in the tables.
In conclusion, it is explained that the application of electrochemical cell-based biosensors in the detection of tumor cells needs to be further explored due to the difficulty of sensing detection in vivo, and the existing electrochemical cell sensing technology has not been widely used in clinical practice.
There are some tipping errors: spaces missing before references (lines 22, 27, 30, 36, 157, 160, 165, 167, 173, 174, 179, 203, 206, 212, etc.) or too many spaces between words (lines 352, 130, 81). Other recommendations are:
- replace figure 4 with clearly one,
- explain the abbreviations from Modification Material columns under each table because there are some that are not explained in the text
- replace “2.4. liver cancer” with “2.4. Liver cancer”
- replace “Hela” (line 210) with “HeLa”
- change “PEDOT-Aunano”(Table 5.) with another more adequate
- replace “Table 44. “(line 106) with CD 44
- correct on bibliography with uppercase letters some abbreviations (12- Pcn-224, 14- Mcf-7, 17, 19- Cd44, 23- Tio 2, 39- Gqds-Concanavalin 518 a@Fe3o4, etc.)
In Table 6 add more Cancer Markers (CA19-9 for example) for gastric cancer with its electrochemical sensors.
The recommendation is to accept after minor revision.
Author Response
Responses to reviewers' comments
We are quite grateful to the kind indications and suggestions from the reviewers. The manuscript has been carefully revised according to the reviewers’ comments. Here are the answers to each of the comments, and the corresponding corrections or change, which have been labeled by red color in the marked revision. I hope my modification can conform to the request of yours.
Reviewer #1:
Comments:
In conclusion, it is explained that the application of electrochemical cell-based biosensors in the detection of tumor cells needs to be further explored due to the difficulty of sensing detection in vivo, and the existing electrochemical cell sensing technology has not been widely used in clinical practice.
There are some tipping errors: spaces missing before references (lines 22, 27, 30, 36, 157, 160, 165, 167, 173, 174, 179, 203, 206, 212, etc.) or too many spaces between words (lines 352, 130, 81).
Responses: Thanks for your scrutiny into our manuscript. We have checked the entire manuscript and corrected it. Spaces have been added (lines 22, 27, 30, 36, 157, 160, 165, 167, 173, 174, 179, 203, 206, 212, etc.) or reduced (lines 352, 130, 81).
Other recommendations are: replace figure 4 with clearly one and explain the abbreviations from Modification Material columns under each table because there are some that are not explained in the text.
Responses: Thanks for your scrutiny into our manuscript. We have replaced figure 4 with clearly one and added the content that are not explained in the text.
Replace “2.4. liver cancer” with “2.4. Liver cancer”, replace “Hela” (line 210) with “HeLa”, change “PEDOT-Aunano”(Table 5.) with another more adequate, replace “Table 44. “(line 106) with CD 44 and correct on bibliography with uppercase letters some abbreviations (12- Pcn-224, 14- Mcf-7, 17, 19- Cd44, 23- Tio 2, 39- Gqds-Concanavalin 518 a@Fe3o4, etc.).
Responses: We have replaced “2.4. liver cancer”, “Hela”, “PEDOT-Aunano” and “Table 44 ” as required. We have checked the references: those abbreviations are corrected on12, 14, 23, 17, 19, 23 and 39.
In Table 6 add more Cancer Markers (CA19-9 for example) for gastric cancer with its electrochemical sensors. The recommendation is to accept after minor revision.
Responses: Thank you for this suggestion to make our manuscript better. We have explored relevant publications for the past five years. There are 5 publications based on electrochemical cell biosensors for detection of gastric cancer. Most of the electrochemical sensors for detection of the gastric cancer markers were conducted in serum or pure antigenic proteins nor the tumor cells. We expanded Table 6 and added all the five literatures into it.
Thank you again for your comments to our manuscript. We make grateful acknowledgement for your careful work. We hope you can give us another opportunity to revise it if there are also some mistakes or questions in our manuscript.
Your comments and suggestions will be highly appreciated.
I am looking forward to hearing from you soon.
Sincerely yours,
Xin Du

Reviewer 2 Report
The authors present a comprehensive review of the literature over the past five years related to various electrochemical cytosensors for cancer detection.
I have the following suggestions for improvements:
- Please check the English grammar carefully. Especially the use of definitive articles, as is for example at row 29.
- The paper contains several sections on different cancers. The reader needs information in the introduction on the content of the paper. I would suggest including in the introduction a scheme depicting a human body on which all cancers reviewed in the paper are marked. This scheme will serve as graphical content.
- The list of literature sources is rather short and it could be expanded with several more titles. Biochip devices are not mentioned although they are closely related to the review topic and deserve a proper reference: Lipid cubic phases as stable nanochannel network structures for protein biochip development: X-ray diffraction study, A Angelova, M Ollivon, A Campitelli, C Bourgaux, Langmuir 19 (17), 6928-6935.
Author Response
Responses to reviewers' comments
We are quite grateful to the kind indications and suggestions from the reviewers. The manuscript has been carefully revised according to the reviewers’ comments. Here are the answers to each of the comments, and the corresponding corrections or change, which have been labeled by red color in the marked revision. I hope my modification can conform to the request of yours.
Reviewer #2:
Comments:
The authors present a comprehensive review of the literature over the past five years related to various electrochemical cytosensors for cancer detection.
I have the following suggestions for improvements:
- Please check the English grammar carefully. Especially the use of definitive articles, as is for example at row 29.
Responses: Thanks for your scrutiny into our manuscript. We have deleted a definitive article at row 29.
- The paper contains several sections on different cancers. The reader needs information in the introduction on the content of the paper. I would suggest including in the introduction a scheme depicting a human body on which all cancers reviewed in the paper are marked. This scheme will serve as graphical content.
Responses: Thank you for this suggestion to make our manuscript better. We have added a figure to describe the distribution of these cancers in the human body.
- The list of literature sources is rather short and it could be expanded with several more titles. Biochip devices are not mentioned although they are closely related to the review topic and deserve a proper reference: Lipid cubic phases as stable nanochannel network structures for protein biochip development: X-ray diffraction study, A Angelova, M Ollivon, A Campitelli, C Bourgaux, Langmuir 19 (17), 6928-6935.
Responses: Thank you for this suggestion to make our manuscript better. We reviewed the last five years of publications on electrochemical cell sensors based on biochips. The biochip is a perfect combination of electrochemical cell sensors. We have discussed related result and citied the literature you recommended in the introduction sections at lines 36-39.
Thank you again for your comments to our manuscript. We make grateful acknowledgement for your careful work. We hope you can give us another opportunity to revise it if there are also some mistakes or questions in our manuscript.
Your comments and suggestions will be highly appreciated.
I am looking forward to hearing from you soon.
Sincerely yours,
Xin Du
